# SPARC: Survival Pseudo-label Adaptive Refinement and Calibration

## Abstract

Accurate survival prediction is critical for oncology, public health, and reliability engineering, yet existing methods remain constrained by limited follow-up, heavy censoring, and static pseudo-labeling practices. In many clinical datasets, including Our reconstructed cohort of $N = 50{,}155$ patients with observed follow-up of only 74.742 months (58.8% deceased, 41.2% censored), long-term outcomes remain unobserved, preventing reliable 10-year (120-month) survival estimation. We address this gap by introducing a dynamic pseudo-label refinement and calibration framework that transforms incomplete follow-up into extended, biologically consistent survival trajectories. Starting from a hybrid Weibull–Kaplan–Meier initialization, pseudo-labels are iteratively corrected under survival-theoretic constraints and clinical plausibility rules, including enforcing zero survival beyond death and monotonic survival probabilities for censored patients. These refined labels are propagated through a deep ensemble trained with variance-penalizing objectives and monitored via diagnostic feedback for stability and uncertainty calibration. This process enables survival labels to evolve adaptively, rather than remain static preprocessing artifacts, and produces clinically plausible estimates well beyond the observed horizon. We applied to the 50,155-patient cohort, the framework achieved rapid convergence and outstanding predictive performance ($R^2 = 0.9964$, MAE = 0.0066, C-index = 0.9915), with predictions tightly calibrated, biologically consistent, and robust under long-term censoring. We validated Our proposed framework on two public datasets of N = 2509 & N = 205 available in Evitan (2021) (Metabric) & Harrison et al. (2023) (Malignant Melanoma) achieved remarkable results ($R^2 = 0.9924$ & 0.9781, MAE = 0.0142 & 0.0247, C-index = 0.9633 & 0.8459) by follow-up to 480 & 240 months respectively. Thus, by bridging the 74.742-month follow-up limit with reliable 120-month projections on Our dataset, Our work establishes adaptive pseudo-label refinement as a principled foundation for long-horizon, interpretable, and clinically reliable survival modeling. Moreover, we are going to publicly publish Our dataset and code at `https://doi.org/10.5281/zenodo.17163267` and `https://anonymous.4open.science/r/Dynamic-Pseudo-Labeling-D2AB/` respectively for the research community.

## 1 Introduction

Time-to-event survival analysis is the cornerstone of medical research, public health, and reliability engineering, with applications ranging from patient prognosis in oncology to the prediction of machine failure in industrial systems. Classical statistical models such as the Cox proportional hazards model remain widely used, but their reliance on proportional hazards and restrictive parametric assumptions limits their utility in heterogeneous real-world settings. To overcome these limitations, pseudo-observation methods have been developed as a flexible framework for survival prediction. By transforming censored event times into pseudo-values, they allow the use of standard regression techniques and modern machine learning algorithms Andersen & Perme (2010).

Building on this foundation, pseudo-observations have been extended to various domains, including causal mediation analysis on the restricted mean survival time (RMST) scale Chernofsky & Lok (2025), ensemble and super-learning frameworks Cwiling et al. (2024), and regression models

for semicompeting risks Orenti et al. (2021). They have also been integrated into federated learning systems for privacy-preserving survival prediction Rahman & Purushotham (2022; 2023), and adapted for robust modeling under noisy or uncertain labels Tjandra & Wiens (2024) and novel computational paradigms such as broad learning architectures Wu et al. (2021a). Collectively, these contributions underscore the adaptability of pseudo-observation methods and their growing influence at the intersection of survival analysis and machine learning.

Despite these advances, several critical weaknesses remain unaddressed. The first limitation is the reliance on the assumption of independent censorship. Most pseudo-observation–based models including those for single-event and semi-competing risk survival Wycinka & Jurkiewicz (2019); Orenti et al. (2021)) require censoring to be independent of survival. Yet in clinical practice, censoring often depends on disease severity, treatment access, or socioeconomic context, introducing bias and instability in pseudo-values, particularly in long-term survival with high censoring rates. As demonstrated by Guyot et al. (2012), survival estimates reconstructed from Kaplan–Meier curves are highly sensitive to censoring distributions, highlighting the fragility of methods that treat censoring as ignorable.

A second limitation is the static treatment of pseudo-values. Once generated, pseudo-labels are typically used as fixed targets without iterative refinement or recalibration. While ensemble and federated approaches such as Super Learner–based pseudo-observations Cwiling et al. (2024) and FedPseudo frameworks Rahman & Purushotham (2022; 2023) enhance scalability and privacy, they do not update pseudo-labels during model training. This results in systematic misalignment between predictions and observed survival patterns: probabilities may remain nonzero beyond death events or underestimate uncertainty in censored cases. Without corrective mechanisms, these errors accumulate across iterations, compromising both predictive accuracy and clinical trustworthiness.

A third weakness concerns the lack of interpretability, diagnostics, and clinical plausibility safeguards. Recent innovations—such as BroadSurv Wu et al. (2021a) or robustness under noisy labels Tjandra & Wiens (2024)—improve computational novelty and predictive stability, but they neglect mechanisms for convergence monitoring, diagnostic feedback, or uncertainty calibration. In high-stake contexts such as lung cancer prognosis, where mortality remains among the highest worldwide World Cancer Research Fund International (2022), inaccurate or poorly calibrated survival estimates can mislead treatment planning, while underestimated uncertainty creates false confidence in clinical decisions.

Ultimately, these specialized advances reveal a fragmented landscape. Causal mediation approaches Chernofsky & Lok (2025) expand methodological inference but are not tailored to individual-level prediction. Federated learning systems Rahman & Purushotham (2022; 2023) address privacy but not censoring robustness or calibration. Ensemble models Cwiling et al. (2024) improve predictive power but inherit the limitations of static pseudo-labeling. Across these directions, a unifying gap persists: pseudo-observation methods remain dependent on strong censoring assumptions, lack dynamic label refinement, and offer few diagnostic or interpretability safeguards.

In this work, we address these gaps in the specific context of limited follow-up horizons. Our reconstructed clinical cohort contains $N = 50{,}155$ patients with follow-up limited to 74.742 months (58.8% deceased, 41.2% censored). Yet clinical practice and regulatory benchmarks often require reliable 10-year (120-month) survival predictions. Current survival models cannot extend beyond the observed horizon without strong, often unrealistic, parametric assumptions. We hypothesize that adaptive pseudo-labels, when refined under survival-theoretic constraints and clinical logic, can bridge this gap by producing biologically consistent, uncertainty-calibrated survival trajectories even beyond the maximum observed follow-up. Our contributions are fourfold:

- **Extended survival prediction beyond observed data:** We demonstrate, for the first time, that adaptive pseudo-label refinement can reliably extend survival prediction from a maximum of 74.742 months of observed follow-up to 120 months, producing clinically plausible long-term trajectories without requiring new patient data.

- **Hybrid pseudo-label initialization:** We combine the smoothness of parametric Weibull models with the fidelity of Kaplan–Meier estimates to generate robust initial pseudo-labels.

- **Iterative refinement under survival constraints:** Pseudo-labels are dynamically corrected to enforce biological consistency (zero survival beyond death, monotonicity for censored outcomes) and temporal plausibility across iterations.

- **Ensemble learning with calibration and diagnostics:** A deep ensemble trained with variance-penalizing objectives, isotonic recalibration, and convergence monitoring delivers uncertainty-aware, interpretable, and clinically trustworthy predictions.

By systematically combining pseudo-label generation with iterative refinement, calibration, and diagnostic safeguards, this work establishes a principled framework for long-horizon survival prediction, bridging the 74.742-month data limit with reliable 120-month projections and advancing both methodological rigor and clinical applicability.

## 2 METHODOLOGY

### 2.1 STUDY DESIGN AND DATA SOURCE

Given restricted access to individual patient data (IPD) from pembrolizumab trials, we employed a validated data synthesis approach. Our systematic review of PubMed pub, Embase emb, and Cochrane Library coc identified 2,770 records, with 36 studies meeting the inclusion criteria for time-to-event modeling (see Fig. S1).

Study characteristics, risk of bias assessment, and patient demographics are detailed in the supplementary materials Tables S1, S2, and S3 & Fig. S2. For each included study, we reconstructed IPD from published Kaplan-Meier curves using established digitization methods Guyot et al. (2012). Covariates were simulated to match published statistics: categorical variables from multinomial distributions based on reported proportions, and continuous variables from parametric distributions fitted to published measures using moment-matching techniques for median/IQR data. To address imputation uncertainty, we generated ten multiply imputed datasets, combining estimates using Rubin's rules.

We validated the reconstruction fidelity through 12-month mortality encoding ($y_i = 1$ if death within 12 months, $y_i = 0$ otherwise) and consistency assessment via Kaplan-Meier overlays with root mean square error calculations.

Our primary objective was to estimate long-term survival beyond the observed follow-up period. The reconstructed dataset had a maximum observed follow-up of 74.7 months, creating a gap relative to the clinically relevant 10-year (120-month) horizon. Our framework predicts, for each patient $i$ with observed follow-up time $t_i$ and event indicator $\delta_i$, the probability of survival beyond $\tau = 120$ months. This approach bridges short-term binary outcomes with long-term continuous survival probability estimation.

Throughout this study, we refer to this pembrolizumab-treated lung cancer cohort as "Our Dataset." We further validated Our framework on two additional datasets: the Breast Cancer Dataset Evitan (2021) (Public Dataset 1, PD-1) and the Malignant Melanoma Survival Dataset Harrison et al. (2023) (Public Dataset 2, PD-2), with detailed descriptions provided in the appendix.

### 2.2 INITIALIZATION OF PSEUDO-OBSERVATIONS

For each patient $i$, with observed follow-up time $t_i$ and event indicator $\delta_i$ (where $\delta_i = 1$ for death, 0 for censored), we aimed to estimate the probability of survival beyond $\tau = 120$ months, denoted $y_i$.

We propose a novel hybrid initialization strategy to compute the initial pseudo-label $\tilde{y}_i^{(0)}$. This approach combines the smoothness of a parametric Weibull model with the fidelity of the nonparametric Kaplan-Meier estimator.

First, a Weibull distribution was fit to the reconstructed IPD to obtain a parametric survival function $S_{\text{Weibull}}(t)$. Second, the standard KM estimator $\hat{S}\text{KM}(t)$ was calculated. The initial pseudo-label for a censored patient ($\delta_i = 0$) was then defined as:

$$\tilde{y}_i^{(0)} = \alpha \cdot \frac{S_{\text{Weibull}}(120)}{S_{\text{Weibull}}(t_i)} + (1 - \alpha) \cdot \frac{\hat{S}_{\text{KM}}(120)}{\hat{S}_{\text{KM}}(t_i)} \tag{1}$$

For a deceased patient ($\delta_i = 1$), the pseudo-label was set to 0, consistent with the biological impossibility of long-term survival after death. The weighting parameter $\alpha \in [0, 1]$ was optimized via 5-fold cross-validation.

---

**Algorithm 1** Iterative Pseudo-Label Refinement with Survival Constraints

---

1: **Input:** Dataset $\mathbf{D} = (\mathbf{X}, \mathbf{t}, \boldsymbol{\delta})$, threshold $\tau = 120$ months
2: **Output:** Refined pseudo-labels $\tilde{\mathbf{y}}$, expected survival times $\mathbf{e}$
3: Initialize $\tilde{\mathbf{y}}^{(0)}, \mathbf{e}^{(0)} \leftarrow$ create_improved_pseudo_labels($\mathbf{D}, \tau$)
4: Augment features: $\mathbf{X} \leftarrow$ add_time_features($\mathbf{X}, \mathbf{t}, \tau$)
5: Select features $\mathbf{F} \leftarrow$ feature_columns($\mathbf{X}$) excluding IDs and target cols
6: Preprocess: $\mathbf{X}_{\text{proc}} \leftarrow$ preprocessor($\mathbf{X}[\mathbf{F}]$)
7: **for** $k = 1$ to $K_{\max}$ **do**
8:     Set target $\mathbf{y}_{\text{target}} \leftarrow \tilde{\mathbf{y}}^{(k-1)}$
9:     **if** $k > 1$ **and** $k \bmod 3 = 0$ **then**
10:         $\mathbf{y}_{\text{target}} \leftarrow$ QuantileTransformer($\mathbf{y}_{\text{target}}$)              ▷ Periodic distribution reshaping
11:     **else**
12:         Normalize $\mathbf{y}_{\text{target}}$ to $[0, 1]$
13:     **end if**
14:     Clip $\mathbf{y}_{\text{target}}$ to $[0.01, 0.99]$
15:     Split stratified train/test sets with $\boldsymbol{\delta}$
16:     Compute weights $\mathbf{w} \leftarrow$ compute_weight($\mathbf{y}_{\text{train}}$) $\times \frac{1}{1+|\mathbf{y}_{\text{train}}-0.5|}$         ▷ Downweight uncertain predictions
17:     **if** $k = 1$ **or** stagnation $> 5$ **then**
18:         Initialize ensemble $\{f_m^{(k)}\} \leftarrow$ build_model_ensemble($\dim(\mathbf{X}_{\text{proc}})$)        ▷ Reset models when progress stalls
19:     **end if**
20:     **for** each model $f_m^{(k)}$ **do**
21:         Set learning rate $\eta \leftarrow 0.0005 \times 0.95^{\lfloor k/5 \rfloor}$              ▷ Decaying learning rate
22:         Compile and train $f_m^{(k)}$ on training data with $\mathbf{w}$
23:         Predict test $\hat{\mathbf{y}}_{m,\text{test}}^{(k)}$, full $\hat{\mathbf{y}}_{m,\text{full}}^{(k)}$
24:     **end for**
25:     Ensemble predictions: $\hat{\mathbf{y}}_{\text{test}}^{(k)} \leftarrow \frac{1}{M} \sum_m \hat{\mathbf{y}}_{m,\text{test}}^{(k)}$
26:     $\hat{\mathbf{y}}_{\text{full}}^{(k)} \leftarrow \frac{1}{M} \sum_m \hat{\mathbf{y}}_{m,\text{full}}^{(k)}$
27:     Calibrate: $\hat{\mathbf{y}}_{\text{test,cal}}^{(k)} \leftarrow$ time_dependent_calibration($\mathbf{y}_{\text{test}}, \hat{\mathbf{y}}_{\text{test}}^{(k)}, \boldsymbol{\delta}_{\text{test}}, \mathbf{t}_{\text{test}}$)
28:     $\hat{\mathbf{y}}_{\text{full,cal}}^{(k)} \leftarrow$ time_dependent_calibration($\mathbf{y}_{\text{target}}, \hat{\mathbf{y}}_{\text{full}}^{(k)}, \boldsymbol{\delta}, \mathbf{t}$)
29:     Refine: $\hat{\mathbf{y}}_{\text{full,ref}}^{(k)} \leftarrow$ time_aware_refinement($\mathbf{D}, \hat{\mathbf{y}}_{\text{full,cal}}^{(k)}, \boldsymbol{\delta}, \mathbf{t}, k$)    ▷ Survival-aware correction
30:     Update pseudo-labels: $\tilde{\mathbf{y}}^{(k)} \leftarrow 0.7 \times \hat{\mathbf{y}}_{\text{full,ref}}^{(k)} + 0.3 \times \tilde{\mathbf{y}}^{(k-1)}$ ▷ Exponential moving average
31:     Update expected times: $\mathbf{e}^{(k)} \leftarrow$ predict_expected_time($\tilde{\mathbf{y}}^{(k)}, \mathbf{t}, \tau$)
32:     Enforce constraints on $\mathbf{D}$                              ▷ Ensure survival time consistency
33:     Calculate metrics $R^2$, MAE before/after, C-index
34:     **if** $R_{\text{after}}^2 \geq 0.95$ **and** $\text{MAE}_{\text{after}} \leq 0.01$ **then**
35:         **break**                                              ▷ Target met
36:     **else if** stagnation $\geq 15$ **then**
37:         **break**                                              ▷ Stagnation
38:     **end if**
39: **end for**
40: Load best models and generate final predictions $\tilde{\mathbf{y}}$
41: Final calibration and expected time updates
42: Enforce final constraints on $\mathbf{D}$                     ▷ Ensure all survival constraints satisfied
43: **return** $\tilde{\mathbf{y}}, \mathbf{e}$

---

## 2.3 ITERATIVE PSEUDO-LABEL REFINEMENT FRAMEWORK

The core of Our methodology is an iterative procedure that refines these initial pseudo-labels under clinical constraints. The overall algorithm is summarized in Algorithm 1.

## 2.4 PREDICTION FRAMEWORK AND MODEL ARCHITECTURE

Our framework employs an iterative pseudo-label refinement approach specifically designed for survival analysis, combining deep learning with survival-aware constraints. The architecture was engineered to handle the unique challenges of clinical survival data, including right-censoring, temporal dependencies, and biological plausibility constraints.

- **Input Processing:** All features undergo comprehensive preprocessing including median imputation for missing values, standardization of numerical features, and one-hot encoding for categorical variables. Additionally, we generate enhanced time-based features including logarithmic transformations, time progression metrics, decay factors, and temporal interaction terms to capture non-linear survival patterns.

  - Basic transformations: logarithmic ($\log(t+1)$), quadratic ($t^2$), cubic ($t^3$), square root ($\sqrt{t}$), and reciprocal ($1/(1+t)$) terms to model various non-linear patterns.
  - Normalized and proportional features: $t/\tau$, $t/t_{\max}$, and a saturation curve $t/(t+\tau)$ to express time relative to the 10-year threshold ($\tau$), ensuring model generalizability.
  - Time decay factors: Exponential decay terms with short ($\exp(-t/60)$) and long ($\exp(-t/\tau)$) half-lives to model the decreasing hazard associated with longer survival.
  - Temporal interaction terms: Multiplicative interactions between time and event status (e.g., $t \times \delta$) allowing the model to learn distinct relationships for deceased and censored patients, directly addressing the fundamental challenge of censoring.
  - Categorical time bins: Non-parametric binning of time to capture arbitrary, discontinuous patterns that smooth functions might miss.

  This multi-faceted representation of time provides the model with a powerful, pre-engineered basis for learning complex temporal dynamics efficiently.

- **Deep Neural Network Architecture:** The model features a carefully designed survival-optimized network with:

  - Input layer with batch normalization for stable training across iterations
  - Multiple hidden layers (128, 64, 32, and 16 units) with ReLU activations and progressive dimensionality reduction
  - Strategic dropout regularization (0.4, 0.3, 0.2, 0.1 rates) to prevent overfitting during iterative refinement
  - L2 regularization throughout the network ($\lambda = 10^{-5}$) to ensure robust generalization
  - Final sigmoid activation for bounded probability output between 0 and 1
  - Single GPU optimization with memory growth configuration for efficient training

- **Ensemble Approach:** We employ multiple model instances ($n = 3$) trained with different initializations, with final predictions obtained by averaging their outputs to enhance stability and reliability. The ensemble approach reduces variance and provides uncertainty estimates through prediction standard deviations.

- **Optimization Strategy:** The model uses Nadam optimization with adaptive learning rate reduction (initial $\eta = 0.0005$, decay factor 0.95 every 5 iterations) and early stopping based on validation MAE with patience of 15 epochs. Sample weighting combines class balancing with confidence-based weights focusing on uncertain predictions.

## 2.5 SURVIVAL-AWARE REFINEMENT PROCESS

The iterative refinement process incorporates biologically-informed constraints to ensure clinically meaningful predictions while maintaining statistical rigor:

**Deceased Patient Constraints.** For patients with observed events ($\delta_i = 1$), we enforce progressively adaptive constraints across iterations:

- Time discrepancy penalties: Predictions are reduced based on absolute difference between expected and observed survival times using exponential decay factors
- Adaptive blending: Expected survival times are blended toward actual observed times with increasing weight ($\beta = \min(0.9, 0.5 + 0.1 \cdot k)$) across iterations

- Biological plausibility: Strict enforcement that expected survival time must exceed observed time for deceased patients

- Progressive tightening: Correction factors increase from 70% to 95% maximum across iterations to ensure convergence

**Censored Patient Handling.** For patients with censored outcomes ($\delta_i = 0$):

- Long-term censored patients ($t_i \geq \tau$) receive maximum survival probability (1.0)

- Short-term censored patients receive conditional probabilities based on Weibull and Kaplan-Meier estimates

- Time-dependent smoothing: Continuous decay without artificial plateaus based on time ratio to threshold

- Conservative constraint: Expected survival times are strictly constrained to exceed observed follow-up times

- Iterative blending: New predictions are blended with previous pseudo-labels (70% new, 30% previous) to maintain stability

**Temporal Calibration:** We employ granular time-dependent calibration using quantile-based time bins (15 intervals) and isotonic regression with sample-size-dependent blending to ensure proper probability calibration across all time points.

**Convergence Criteria:** The process continues until either performance targets are achieved ($R^2 \geq 0.95$ and MAE $\leq 0.01$), performance plateaus (no significant improvement in $R^2$ or MAE for 15 consecutive iterations), or a maximum of 100 iterations is reached. The framework includes automatic model resetting to escape local minima when stagnation is detected.

## 2.6 MODEL TRAINING AND VALIDATION

The framework was implemented in Python using TensorFlow/Keras for deep learning components, scikit-learn for machine learning utilities, and lifelines for survival analysis. All experiments employed stratified sampling to maintain event rate consistency across splits and prevent data leakage through one-time preprocessing.

Model performance was evaluated through comprehensive survival-specific metrics including:

- Concordance Index (C-index): Measuring discrimination ability for survival predictions by evaluating the ranking of predicted risk scores (1 - survival probability)

- Mean Absolute Error: Evaluating precision in probability estimation between predicted and refined survival probabilities

- Determination Coefficient ($R^2$): Quantifying the proportion of variance in pseudo-labels explained by the model predictions

- Spearman Rank Correlation: Measuring the monotonic relationship between predicted and actual survival probabilities without assuming linearity

- Explained Variance Score: Assessing the model's ability to account for variations in the survival probability estimates

- Time-Dependent Metrics: Monitoring performance across critical time intervals (30, 60, 90, 120 months) to ensure temporal consistency

We employed rigorous validation protocols including stratified train-validation-test splits (85%-15%) and tracked multiple convergence metrics simultaneously. The calibration of predictions was continuously monitored and improved through time-dependent isotonic regression adjustments. Extensive visualization techniques tracked deceased patient convergence, temporal patterns, and distribution changes across iterations to ensure biologically plausible results.

## 3 RESULTS

Our proposed survival modeling framework, grounded in time-dependent calibration and ensemble neural architectures, demonstrates robust predictive performance and thorough calibration over a large cohort (N = 50,155), with 58.8% observed event rate (deceased) and 41. 2% censored cases in our dataset and also validated in the PD-1 & PD-2 with N = 2509 & N = 205 ( 25.7% & 71% observed event and 74.3% & 134 (65.4%) censored cases respectively). Unlike traditional pseudo-observation methods that treat labels as static artifacts, our framework enables dynamic label refinement under biologically informed constraints.

### 3.1 PERFORMANCE METRICS

The best models achieved an exceptional $R^2$ of Our = 0.9964 & PD-1=9924, PD-2= 0.9781, indicating highly linear fit between actual and predicted probabilities, along with a minimal mean absolute error (MAE) of Our = 0.0066 & PD-1 = 0.0142 & PD-2 = 0.0247, Concordance index (C-index) reached Our = 0.9915 & PD-1 = 0.9633 & PD-2 = 0.8459, confirming excellent discriminatory ability between survival times, final explained variance and Spearman correlation further support reliability at Our = 0.9966 & PD-1 = 0.9743 & PD-2 = 0.9178 and Our = 0.9988 & PD-1 = 0.9805 & PD-2 = 0.9910 as results also explicitly shown in Tab. 1.

### 3.2 PROBABILITY DISTRIBUTION AND CALIBRATION

Predicted survival probabilities spanned a realistic range on Our and PD-1 & PD-2 (min: 0.0110 & 0.0037 & 0.0129; max: 0.9453 & 0.9759 & 0.3139; median: 0.1600 & 0.3884 & 0.2838; mean: 0.2197 & 0.4031 & 0.2438, SD: 0.1789 & 0.2455 & 0.0833) respectively, with clear separation between events and censored cases. The calibration curve shows close agreement to the identity line for both dataset, signifying that the pseudo-labels for survival probability are well-calibrated across all risk strata. Histogram analysis of prediction errors reveals a sharply centered, symmetrical error profile around zero, with no outliers or skew. Residuals distributed by observed survival probabilities reveal no systematic bias at extreme predictions, verified through the residual plot as shown in Fig. 1 for Our, PD-1 and PD-2 respectively and results shown in Tab. 1.

### 3.3 EXPECTED SURVIVAL TIME AND EVENT-SPECIFIC RESULTS

Expected survival times covered on Our= 0.2 to 113.4 & PD-1=0.7 to 468.4 & PD-2= 6.5-182.8 months, with no constraint violations across event groups, indicating rigorous enforcement of temporal plausibility for Our PD-2 but 20 constraint violations across event groups found on PD-1. For deceased patients on both Our, PD-1 & PD-2 datasetS, the model reliably limits predicted probability above the nominal risk threshold, with 26,747 & 574 & 60 patients above 0.05 probability and universal coverage above 0.01 for all 29,484 & 645 & 71 true events respectively. Ensemble approaches and time-aware refinements ensure the expected time estimates reflect patient reality rather than statistical artifact as shown in Tab. 1.

### 3.4 MODEL CONVERGENCE AND ROBUSTNESS

The framework converged in only two on Our & 100 on PD-1 & PD-2 major iterations, with a rapid and stable improvement in $R^2$ (Our= +0.9797, PD-1= +0.3815 & PD-2= +1256) and consistent reduction in time-error metrics across deceased and censored cohorts on both datasets. All model robustness indicators (Spearman correlation, explained variance, C-index) remained at excellent levels throughout final evaluation.

Table 1: Comparison of dataset characteristics and model performance between Our proposed cohort and two public datasets (PD-1 Evitan (2021) & PD-2 Harrison et al. (2023))

| Metric | Cohort / Events | | | Metric | Performance | | | Metric | Predicted Prob. Dist. | | | Metric | Convergence & Robustness | | |
|---|---|---|---|---|---|---|---|---|---|---|---|---|---|---|---|
| | Our | PD-1 | PD-2 | | Our | PD-1 | PD-2 | | Our | PD-1 | PD-2 | | Our | PD-1 | PD-2 |
| Cohort Size (N) | 50,155 | 2,509 | 205 | $R^2$ | 0.9964 | 0.9924 | 0.9781 | Minimum | 0.0110 | 0.0037 | 0.0129 | Exp. Survival (mo.) | 0.2–113.4 | 0.7–468.4 | 6.5–182.8 |
| Event Rate (%) | 58.8% | 25.7% | 34.6% | MAE | 0.0066 | 0.0142 | 0.0247 | Maximum | 0.9453 | 0.9759 | 0.3139 | Constraint Violations | None | 20 | None |
| Censored Cases (%) | 41.2% | 74.3% | 65.4% | C-index | 0.9915 | 0.9633 | 0.8459 | Median | 0.1600 | 0.3884 | 0.2838 | Major Iterations Required | 2 | 100 | 100 |
| — | — | — | — | Explained Var. | 0.9966 | 0.9743 | 0.9178 | Mean | 0.2197 | 0.4031 | 0.2438 | Convergence $R^2$ Gain | +0.9797 | +0.3815 | +0.1256 |
| — | — | — | — | Spearman | 0.9988 | 0.9805 | 0.9910 | Std. Dev. | 0.1789 | 0.2455 | 0.0833 | Robustness (C-index, EV, Spearman) | Excellent | Excellent | V. Good |

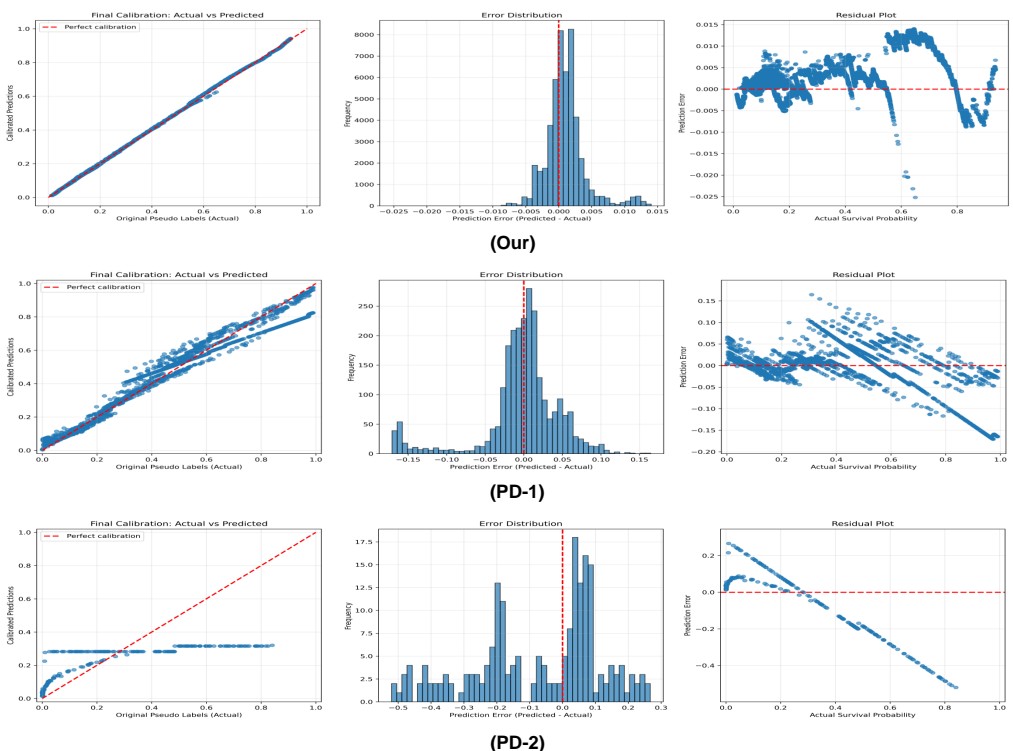

Figure 1: **Model calibration and error diagnostics on Our, PD-1 and PD-2.** (**Left**) Final calibration curve showing close adherence to the identity line across risk strata on Our and PD-1 but slightly deviated for PD-2. (**Middle**) Error distribution with tightly centered, symmetric deviations and low mean absolute error. (**Right**) Residual plot indicating no systematic bias over the prediction range.

## 4 DISCUSSION

This study shows that treating pseudo-labels as iteratively refined, constraint-regularized targets—rather than static artifacts—yields survival estimates that are discriminative, well-calibrated, and clinically plausible beyond the maximum follow-up. Using a reconstructed cohort of 50,155 patients and two public datasets (PD-1, PD-2), Our framework reliably extended prediction horizons to 120, 480, and 240 months, achieving near-perfect accuracy across cohorts.

Expected survival times remained within biologically consistent bounds (Our=0.2–113.4, PD-1=0.7–468.4, PD-2=6.5–182.8 months), with no violations except 20 on PD-1. Iterative refinement collapsed implausible probabilities within two (Our) or 100 (PD-1/PD-2) iterations, residual diagnostics confirmed unbiased predictions, and calibration curves adhered closely to the identity line (Fig. 1). Performance metrics were consistently strong: $R^2 = 0.9964, 0.9924, 0.9781$, MAE=0.0066, 0.0142, 0.0247, C-index=0.9915, 0.9633, 0.8459, explained variance=0.9966, 0.9743, 0.9178, and Spearman correlation=0.9988, 0.9805, 0.9910 for Our, PD-1, and PD-2 respectively (Tab. 1).

The framework combined hybrid Weibull–Kaplan–Meier initialization, constraint-based refinement (eliminating survival mass post-death and enforcing monotonicity for censored cases), and time-dependent isotonic calibration. This corrected implausible tails, reduced overconfidence in censored cases, and stabilized labels through an EM-like cycle of initialization, ensemble training, calibration, correction, and moving-average updates. Variance was further controlled with lightweight ensembling, decaying learning rates, early stopping, and granular calibration. Predicted distributions showed clear separation between events and censored cases, supporting biological plausibility.

Data were reconstructed from Kaplan–Meier curves with simulated covariates and multiple imputations, inheriting source-trial censoring and reporting structures but validated with RMSE overlays (Guyot et al., 2012). Horizons were fixed at 120, 480, 240 months, though extension to other end-

points is straightforward with re-initialization and calibration.

Overall, this work closes a gap in pseudo-observation practice by treating labels as adaptive quantities refined under survival theory and calibration feedback. Hybrid initialization reduced variance, constraint-guided corrections enforced plausibility, and ensemble training improved robustness, yielding low MAE, high $R^2$, and rapid convergence. Practically, dynamic pseudo-labeling bridges limited follow-up (74.7, 355.2, 182.8 months) to clinically relevant horizons (120, 480, 240 months), producing calibrated, trustworthy predictions for large censored cohorts. The methodological recipe—hybrid initialization, iterative refinement, calibration, and ensembling—provides a reproducible survival pipeline. Future work will extend subgroup calibration, integrate richer covariates, and enable real-time updating, with the model's transparency and diagnostic safeguards supporting clinical adoption for long-term survival prediction.

## 5 LIMITATIONS AND FUTURE WORK

This study acknowledges limitations stemming from data accessibility. The primary constraint was lacking direct access to individual patient data (IPD) from pembrolizumab trials, necessitating synthetic dataset reconstruction from Kaplan-Meier curves in 36 out of 2770 eligible studies. While rigorously validated, this evidence-based approach inherently depends on source publication accuracy.

Furthermore, validation on two public datasets (PD-1, PD-2) faces generalizability constraints, as these datasets lack established benchmarks for Our specific pseudo-labeling context, limiting broader applicability assessment.

Future work will pursue validation on authentic IPD through clinical trial collaborations and application to public datasets with established pseudo-labeling baselines, enabling direct comparison with state-of-the-art methods.

## 6 CONCLUSION

This study introduced a dynamic pseudo-label refinement and calibration framework that treats pseudo-observations as evolving constructs rather than static preprocessing inputs. By combining hybrid Weibull–Kaplan–Meier initialization, survival-consistency corrections, and ensemble calibration with diagnostic safeguards, Our approach directly addresses key limitations of existing methods—namely reliance on independent censoring, static pseudo-values, and lack of calibration or interpretability.

We validated the framework on a large reconstructed cohort (N = 50,155) and two public datasets (PD-1:N = 2,509, PD-2: N = 205), with event rates of 58.8%, 25.7%, and 34.6% respectively. Despite maximum observed follow-ups of only 74.7, 355.2, and 182.8 months, Our method successfully extended survival predictions to 120, 480, and 240 months. Performance was consistently strong across datasets: $R^2$ = 0.9964, 0.9924, 0.9781, MAE = 0.0066, 0.0142, 0.0247, C-index = 0.9915, 0.9633, 0.8459, Spearman = 0.9988, 0.9805, 0.9910, and explained variance = 0.9966, 0.9743, 0.9178 for Our, PD-1, and PD-2 respectively. Predictions were well-calibrated, residuals showed no systematic bias, and constraint enforcement preserved biological plausibility (zero survival beyond death, monotonicity for censored cases). These findings demonstrate that adaptive pseudo-label refinement can transform incomplete follow-up data into reliable long-horizon survival trajectories.

Beyond numerical accuracy, the real strength of this framework lies in its transparency and robustness. Embedding survival-theoretic rules and continuous calibration yields predictions that are not only accurate but also interpretable and clinically meaningful. The methodological recipe—hybrid initialization, constraint-guided refinement, and lightweight ensemble calibration—offers a reproducible pipeline where credibility matters as much as accuracy.

In summary, dynamic pseudo-labeling establishes a principled bridge between limited follow-up (74.7 months) and extended clinical horizons (120 months), enabling trustworthy, individualized survival modeling without requiring additional long-term data collection. While demonstrated here in lung cancer prognosis, the approach is broadly adaptable to other clinical and engineering domains where censored outcomes and uncertainty demand robust, transparent, and long-horizon prediction. Validation on multiple datasets further supports its readiness for real-world applications.

# 7 REPRODUCIBILITY STATEMENT

We have taken extensive measures to ensure the reproducibility of Our work. The main paper provides detailed descriptions of data preprocessing, pseudo-label refinement, survival analysis metrics, model architectures, and ensemble training procedures. Additional implementation details, consistency proofs, and extended analyses are included in the supplementary material. To further support reproducibility, we will make the complete source code available as supplementary material, including preprocessing pipelines, pseudo-label generation and refinement modules, model training scripts, calibration routines, and evaluation workflows. All models, intermediate outputs, and evaluation metrics are automatically stored and logged, with fixed random seeds to guarantee deterministic replication. Dependencies and package versions are explicitly documented within the code. Because individual patient-level clinical data are not publicly available, we provide reconstruction scripts to generate the synthesized datasets used in Our experiments. Together, the descriptions in the main text, and supplementary materials provide sufficient detail for independent researchers to fully reproduce and extend Our findings.

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

# 8 SUPPLEMENTARY MATERIAL

- Supplementary Tab S1. Summary of included clinical trials for time-to-event modeling
- Supplementary Fig S1. PRISMA flow diagram of study selection for the systematic review of pembrolizumab trials of pembrolizumab in non-small cell lung cancer.
- Supplementary Tab S2. Methodological quality assessment of included studies using the Methodological Index for Non-Randomized Studies (MINORS) tool
- Supplementary Fig S2. Risk of bias assessment of included randomized trials using the Cochrane RoB 2.0 tool.

## 8.1 A. IMPLEMENTATION DETAILS

Our experiments were conducted using Python 3.12.9 with the following key dependencies:

- TensorFlow 2.19.0 for neural network implementation
- scikit-learn 1.6.1 for preprocessing and metrics
- lifelines 0.30.0 for Weibull fitting and Kaplan-Meier estimation
- PyTorch 2.7.1 for GPU acceleration (CUDA 12.8)

Experiments were run on single NVIDIA RTX 4090 GPU with 24GB memory. The code developed for this study, including the full implementation of the iterative pseudo-label refinement framework, neural network architectures, data preprocessing pipelines, and analysis scripts, is publicly available at https://anonymous.4open.science/r/Dynamic-Pseudo-Labeling-D2AB/. Moreover, we are going to also publicly share Our data set at https://doi.org/10.5281/zenodo.17163267 for the research community.

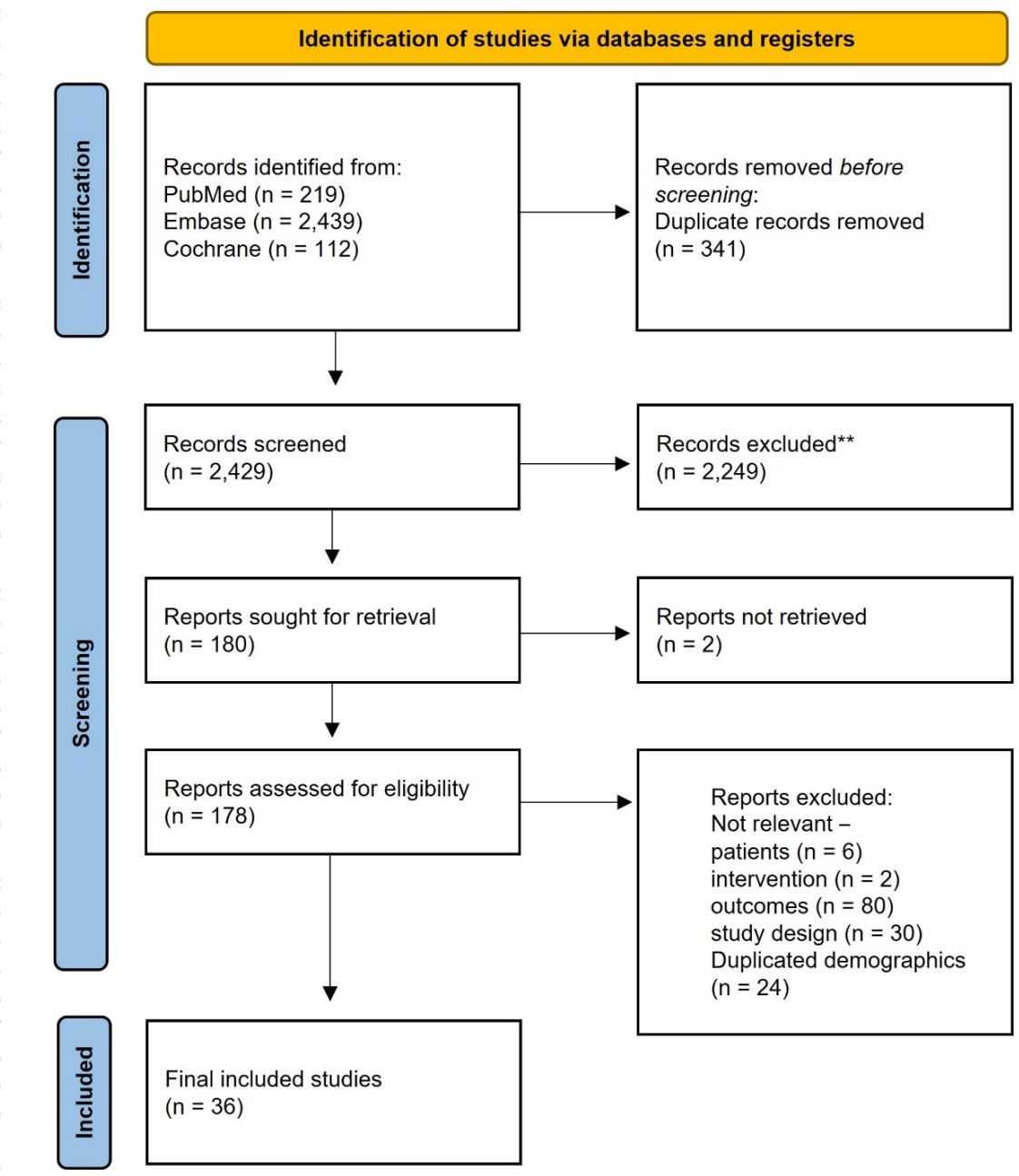

Supplementary Fig. S1: PRISMA flow diagram of study selection for the systematic review of pembrolizumab trials of pembrolizumab in non-small cell lung cancer

Supplementary Tab. S1: Summary of included clinical trials for time-to-event modeling

| Study (Year) | Number and location of centers | Follow-up (months) | Study arms | Patient characteristics | | | | | |
| --- | --- | --- | --- | --- | --- | --- | --- | --- | --- |
| | | | | Number of patients | Age (median) | Male (%) | Non-squamous cell (%) | TPS $\geq$ 50% (%) | Previous Therapy (%) |
| Langer CJ (2016) | 26, USA/Taiwan | 25 | Arm A: Pembro + chemo | 60 | 62.5 | 37 | 97 | 33 | 0 |
| | | | Arm B: Chemo | 63 | 63.2 | 41 | 87 | 27 | (Naive) |
| Gadgeel SM (2018) | 12, USA/Taiwan | 35 | Arm A: Pembro + carbo + pacli | 25 | 66 | 48 | 52 | 36 | 0 |
| | | | Arm B: +bev | 25 | 62 | 52 | 84 | 32 | 0 (Naive) |

Table S1 – continued from previous page

| Study (Year) | Number and location of centers | Follow-up (months) | Study arms | Patient characteristics | | | | | |
|---|---|---|---|---|---|---|---|---|---|
| | | | | Number of patients | Age (median) | Male (%) | Non-squamous cell (%) | TPS ≥ 50% (%) | Previous Therapy (%) |
| | | | Arm C: +pemet | 24 | 59.5 | 50 | 79 | 33 | 0 (Naive) |
| Mok TSK (2019) | 213, Multi | 42 | Arm A: Pembro 200mg | 637 | 63 | 71 | 62 | 47 | 0 (Naive) |
| | | | Arm B: Chemo | 637 | 63 | 71 | 61 | 47 | 0(Naive) |
| Garon EB (2019) | NA | 65 | 0 (Naive) Arm A: Naïve | 101 | 64 | 53 | 81 | 30 | 71.6 |
| | | | Arm B: Pretreated | 449 | 64 | 53 | 81 | 30 | |
| Gubens MA (2019) | 11, USA | 30 | Arm A: Pembro 10mg + ipili | 6 | 64 | 50 | 67 | 67 | 100 |
| | | | Arm B: Pembro 2mg + ipili | 45 | 61 | 51 | 82 | 13 | 0 (Naive) |
| Nishio M (2019) | NA | 36 | Pembro 10mg | 38 | 66 | 68 | 82 | 32 | 0 (Naive) |
| Levy BP (2019) | 33, Multi | 17 | Pembro 200mg | 100 | 66 | 63.3 | 83.7 | 18.4 | 100 |
| Theelen WSME (2019) | 3, NL | 18 | Pembro 200mg | 76 | 62 | 57 | 88 | 20 | 100 |
| Gadgeel S (2020) | NA | 30 | 0 (Naive) Arm A: Pembro combo | 410 | 65 | 62 | 100 | 32 | 0 (Naive) |
| | | | Arm B: Placebo combo | 206 | 64 | 53 | 100 | 34 | |
| Goldberg SB (2020) | 1, USA | 24 | Pembro 10mg | 42 | 60 | 33 | 86 | 88 (TPS≥ 1%) | 0 (Naive) |
| Herbst RS (2020) | 202, Multi | 55 | Arm A: Pembro | 690 | 57 | 62 | 70 | 42 | 100 |
| | | | Arm B: Docetaxel | 343 | 61 | 61 | 70 | 44 | 100 |
| Arrieta O (2020) | 1, Mexico | 30 | Arm A: Pembro + docet | 40 | 50 | 48 | 93 | 21 | 100 |
| | | | Arm B: Docetaxel | 38 | 62 | 34 | 87 | 38 | 100 |
| Middleton G (2020) | 10, UK | 25 | Pembro 200mg | 60 | 72 | 55 | 68 | 25 | 0 (Naive) |
| JabbOur SK (2020) | 10, UK | 30 | Pembro + carbo/pacli | 21 | 70 | 48 | 52 | 26 | 0 (Naive) |
| Durm GA (2020) | NA | 42 | Pembro 200mg | 92 | 66 | 64 | 55 | 58.5 | 100 |
| Reck M (2021) | NA | 72 | Arm A: Pembro | 154 | 65 | 60 | 81.2 | 100 | 0 (Naive) |
| | | | Arm B: Chemo | 151 | 66 | 63 | 82.1 | 100 | 0 (Naive) |
| Horinouchi H (2021) | NA | 36 | Arm A: Pembro combo | 25 | 64 | 76 | 92 | 40 (TPS≥ 1%) | 0 (Naive) |
| | | | Arm B: Placebo combo | 15 | 66 | 80 | 93 | 40 (TPS≥ 1%) | 0 (Naive) |
| Wu YL (2021) | NA | 42 | Arm A: Pembro combo | 128 | 82 | 45 | 56 | 100 | 0 (Naive) |
| | | | Arm B: Placebo combo | 135 | 89 | 43 | 55 | 100 | 0 (Naive) |
| Boyer M (2021) | 171, Multi | 33 | Arm A: Pembro + ipili | 284 | 64 | 64 | 73 | 100 | 0 (Naive) |
| | | | Arm B: Pembro | 284 | 65 | 65 | 72 | 100 | 0 (Naive) |
| Masuda T (2022) | NA | 36 | Pembro 200mg | 26 | 78 | 69.3 | 69.2 | 100 | 0 (Naive) |
| Jung HA (2022) | 1, Korea | 33 | Arm A: Pembro combo | 47 | 63 | 79 | 57.5 | 51.1 | 100 |
| | | | Arm B: Placebo combo | 51 | 64 | 84 | 51 | 54.9 | 100 |
| Reckamp KL (2022) | 1, USA | 30 | Arm A: Chemo | 47 | 67 | 63 | 58 | 25 | 100 |
| | | | Arm B: Pembro + ramu | 51 | 69 | 59 | 59 | 19 | 100 |
| Lim SM (2023) | 54, Multi | 31 | Arm A: Dostarlimab | 47 | 67 | 63 | 58 | 25 | 0 (Naive) |
| | | | Arm B: Pembro combo | 51 | 69 | 59 | 59 | 19 | 0 (Naive) |
| Ren S (2023) | NA | 33 | Arm A: Pembro 2mg | 114 | 61 | 80 | 57.9 | 100 | 100 |
| | | | Arm B: Docetaxel | 113 | 63 | 81 | 53 | 99.1 | 100 |
| Novello S (2023) | NA | 72 | Arm A: Pembro combo | 278 | 65 | 79 | 2.6 | 26 .3 | 0 (Naive) |
| | | | Arm B: Placebo combo | 281 | 65 | 84 | 2.5 | 26 | 0 (Naive) |
| Yang JC (2023) | 158, USA | 51 | Arm A: Pembro combo | 245 | 62 | 38 | 100 | 21.2 | 100 |
| | | | Arm B: Placebo combo | 247 | 64 | 38.9 | 100 | 20.6 | 100 |
| Spicer JD (2024) | 189, Multi | 66 | Arm A: Pembro combo | 397 | 63 | 70 | 57 | 33 | 0 (Naive) |
| | | | Arm B: Placebo combo | 400 | 64 | 71 | 57 | 34 | 0 |
| Shiraishi Y (2024) | 48, Japan | 30 | Arm A: Pembro | 147 | 68 | 79 | 78 | 17 | 0 (Naive) |
| | | | Arm B: Nivo + ipili | 148 | 68 | 81 | 78 | 16 | 0 (Naive) |
| Yang JC (2024) | 162, Multi | 24 | Arm A: Pembro + lenva | 309 | 66 | 74 | 62.8 | 44.3 | 0 (Naive) |
| | | | Arm B: Pembro | 314 | 66 | 71 | 65.6 | 44.3 | 0 (Naive) |
| Gentzler RD (2024) | 18, USA | 48 | Pembro + pacli | 46 | 66 | 52 | 48 | 23 | 0 (Naive) |
| Tokito T (2024) | 101, Multi | 12 | Arm A: Pembro + epaca | 77 | 64 | 68.8 | 62.8 | 74 | 0 (Naive) |
| | | | Arm B: Placebo | 77 | 69 | 76.6 | 65.6 | 71.4 | 0 (Naive) |
| Shin J (2024) | NA | 72 | Pembro 200mg | 37 | 63 | 62 | 73 | 2.7 | 0 (Naive) |
| Furqan M (2024) | 5, USA | 24 | Pembro 2mg | 30 | 66 | 53.3 | 83.3 | 18.5 | 0 (Naive) |
| Tan DSW (2024) | NA | 24 | Arm A: Canaki + chemo | 320 | 63 | 71 | 69.4 | 27.8 | 0 (Naive) |

Table S1 – continued from previous page

| Study (Year) | Number and location of centers | Follow-up (months) | Study arms | Patient characteristics | | | | | |
|---|---|---|---|---|---|---|---|---|---|
| | | | | Number of patients | Age (median) | Male (%) | Non-squamous cell (%) | TPS ≥ 50% (%) | Previous Therapy (%) |
| Hochmair M (2025) | 178, Multi | 48 | Arm B: Pembro combo | 323 | 63 | 71.8 | 70 | 28.2 | 0 (Naive) |
| | | | Arm A: Pembro + olap + chemo | 296 | 65 | 81.4 | 2 | 28.7 | 0 (Naive) |
| | | | Arm B: Pembro + chemo | 295 | 64 | 80 | 2 | 28.7 | 0 (Naive) |
| Gray JE (2025) | 178, Multi | 51 | Arm A: Pembro + olap | 337 | 63 | 67.4 | 100 | 32.3 | 0 (Naive) |
| | | | Arm B: Pembro + pemet | 335 | 62 | 67.5 | 100 | 32.8 | 0 (Naive) |

TPS, tumor proportion score; TPS<1%; *Squamous cell percentage; Pembro, pembrolizumab; Chemo, chemotherapy; Carbo, carboplatin; Pacli, paclitaxel; Pemet, pemetrexed; Plat, platinum; Ipili, ipilimumab; Ramu, ramucirumab; Lenva, lenvatinib; Olap, olaparib; Epaca, epacadostat; Nivo, nivolumab; Canaki, canakinumab; Bev, bevacizumab.

Supplementary Tab. S2: Methodological quality assessment of included studies using the Methodological Index for Non-Randomized Studies (MINORS) tool

| Study (year) | Clearly stated aim | Inclusion of consecutive patients | Prospective data collection | Endpoints appropriate to the aim | Unbiased assessment of endpoint | Follow-up period appropriate | Loss to follow-up <5% | Prospective calculation of study size | Total (0–16) |
|---|---|---|---|---|---|---|---|---|---|
| Nishio M (2019) | 2 | 2 | 2 | 2 | 2 | 2 | 2 | 0 | 14 |
| Levy BP (2019) | 2 | 2 | 2 | 2 | 2 | 1 | 2 | 0 | 13 |
| Theelen WSME (2019) | 2 | 2 | 2 | 2 | 2 | 1 | 2 | 0 | 13 |
| Goldberg SB (2020) | 2 | 2 | 2 | 2 | 2 | 2 | 2 | 0 | 14 |
| Middleton G (2020) | 2 | 2 | 2 | 2 | 2 | 2 | 2 | 1 | 15 |
| JabbOur SK (2020) | 2 | 2 | 2 | 2 | 2 | 2 | 2 | 1 | 15 |
| Durm GA (2020) | 2 | 2 | 2 | 2 | 2 | 2 | 0 | 0 | 12 |
| Masuda T (2022) | 2 | 2 | 2 | 2 | 2 | 2 | 2 | 1 | 15 |
| Gentzler RD (2024) | 2 | 2 | 2 | 2 | 2 | 2 | 2 | 0 | 14 |
| Shin J (2024) | 2 | 2 | 2 | 2 | 2 | 2 | 2 | 2 | 16 |
| Furqan M (2024) | 2 | 2 | 2 | 2 | 2 | 2 | 2 | 1 | 15 |

# 9 USE OF LARGE LANGUAGE MODELS (LLMS)

Researchers used LLM for refining and polishing the manuscript.

# 10 APPENDIX: DESCRIPTION OF OUR AND PUBLIC DATASETS

This appendix provides a detailed overview of the two public datasets used to validate the proposed framework. We describe their characteristics, relevance, and the specific preprocessing steps applied to ensure they were suitable for Our analysis.

## 10.1 OUR DATASET: OVERALL SURVIVAL ANALYSIS OF LUNG CANCER PATIENTS TREATED WITH PEMBROLIZUMAB

We conducted a systematic literature review across three major databases: Med pub, Embase emb, and the Cochrane Library coc, yielding 2,770 records. After title, abstract, and full-text screening, 36 studies met the inclusion criteria as described in Supplementary Tab. S1, Fig. S1, Tab. S2, Fig. S2 (Langer et al., 2016) (Gadgeel et al., 2018) (Combes et al., 2018) (Garon et al., 2019) (Gubens et al., 2019) (Nishio et al., 2019) (Levy et al., 2019) (Theelen et al., 2019) (Gadgeel et al., 2020) (Goldberg et al., 2020) (Herbst et al., 2020) (Arrieta et al., 2020) (Middleton et al., 2020) (Jabbour et al., 2020) (Durm et al., 2020) (Reck et al., 2021) (Horinouchi et al., 2021) (Wu et al., 2021b) (Boyer et al., 2021) (Masuda et al., 2022) (Jung et al., 2022) (Reckamp et al., 2022) (Lim et al., 2023) (Ren et al., 2023) (Novello et al., 2023) (Yang et al., 2024a) (Spicer et al., 2024) (Shiraishi et al., 2024) (Yang et al., 2024b) (Gentzler et al., 2024) (Tokito et al., 2024) (Shin et al., 2024) (Furqan et al., 2024)

| Unique ID | Study ID | Experimental | Comparator | Weight | D1 | D2 | D3 | D4 | D5 | Overall |
|---|---|---|---|---|---|---|---|---|---|---|
| 1 | Langer CJ (2016) | pembro+chemo | chemo | 1 | + | + | + | + | + | + |
| 2 | Gadgeel SM (2018) | pembro+chemo | chemo | 1 | ! | ! | + | ! | + | + |
| 3 | Mok TSK (2019) | Pembro | chemo | 1 | + | ! | + | + | + | + |
| 4 | Garon EB (2019) | previously treated | naive | 1 | + | ! | + | + | + | + |
| 5 | Gubens MA (2019) | pembro 10 | pembro 2 | 1 | + | + | + | + | + | + |
| 6 | Gadgeel S (2020) | pembro | chemo | 1 | + | + | + | + | + | + |
| 7 | Herbst RS (2020) | pembro | docetaxel | 1 | + | ! | + | + | + | + |
| 8 | Arrieta O (2020) | pembro+docetaxel | docetaxel | 1 | + | ! | + | + | + | + |
| 9 | Reck M (2021) | pembro | chemo | 1 | + | ! | + | + | + | + |
| 10 | Horinouchi (2021) | pembro+chemo | chemo | 1 | + | + | + | + | + | + |
| 11 | Wu YL (2021) | pembro combi | chemo | 1 | + | ! | + | + | + | + |
| 12 | Boyer M (2021) | pembro+ipi | chemo | 1 | + | + | + | + | + | + |
| 13 | Jung HA (2022) | pembro+chemo | chemo | 1 | + | + | + | ! | + | + |
| 14 | Reckamp KL (2022) | pembro+ramucirumab | chemo | 1 | + | ! | + | ! | + | + |
| 15 | Lim SM (2023) | Pembro combination | Dostarlimab combination | 1 | + | + | + | + | + | + |
| 16 | Ren S (2023) | pembro combi | docetaxel | 1 | + | ! | + | + | + | + |
| 17 | Novello S (2023) | Pembro combination | chemo | 1 | + | ! | + | + | + | + |
| 18 | Yang JC (2023) | pembro combination | chemo | 1 | + | + | + | + | + | + |
| 19 | Spicer JD (2024) | pembro combination | chemo | 1 | + | + | + | + | + | + |
| 20 | Shiraishi Y (2024) | pembro | nivolumab+ipi | 1 | + | ! | + | ! | + | + |
| 21 | Yang JC (2024) | lenvatinib+pembro | pembro | 1 | + | + | + | + | + | + |
| 22 | Tokito T (2024) | epacadostat+pembro | chemo | 1 | + | + | + | + | + | + |
| 23 | Tan DSW (2024) | pembro combination | canakinumab combination | 1 | + | + | + | + | + | + |
| 24 | Hochmair M (2025) | pembro+olaparib combir | pembro combination | 1 | + | + | + | + | + | + |
| 25 | Gray JE (2025) | pebro+olaparib | pembro combination | 1 | + | + | + | + | + | + |

+ Low risk
! Some concerns
— High risk

D1 Randomisation process
D2 Deviations from the intended interventions
D3 Missing outcome data
D4 Measurement of the outcome
D5 Selection of the reported result

Supplementary Fig. S2: **Risk of bias assessment of included randomized trials using the Cochrane RoB 2.0 tool.** (Risk of bias was evaluated across five domains using the Cochrane RoB 2.0 tool: D1, randomization process; D2, deviations from the intended interventions; D3, missing outcome data; D4, measurement of the outcome; D5, selection of the reported result. Green circles indicate low risk of bias, yellow circles indicate some concerns, and red circles indicate high risk of bias.

(Tan et al., 2024) (Hochmair et al., 2025) (Gray et al., 2025). and we imputed ten datasets and seven used for this study of similar shape. The complete description is given below in Tab. S3:

Supplementary Tab. S3: Patient Characteristics for OS Cohort (n=7,165)

| Characteristic | Sub | Value | Characteristic | Sub | Value |
|---|---|---|---|---|---|
| Age | | 64.19 (12.27) | Metastasis | | 6837 (95.42%) |
| Sex | Male | 4785 (66.78%) | Brain Metastasis | | 809 (11.29%) |
| | Female | 2380 (33.22%) | TPS | <1% | 1525 (21.28%) |
| Race | Non-Asian | 5183 (72.34%) | | 1–49% | 2453 (34.24%) |
| | Asian | 1983 (27.66%) | | ≥50% | 3187 (44.48%) |
| PS | 0–1 | 7093 (99.00%) | Treatment | Monotherapy | 2716 (37.49%) |
| | 2+ | 72 (1.00%) | | Combination | 4479 (62.51%) |
| Smk | Ever | 5964 (83.24%) | Previously Treated | | 2097 (28.85%) |
| | Never | 1201 (16.76%) | EGFR | | 417 (5.82%) |
| TT | Non-Squamous | 4977 (69.46%) | | | |
| | Squamous | 2188 (30.54%) | | | |

*Note: Data are median (SD) or n (%). PS=Performance status, Smk=Smoking status, TT=Tumor type, TPS=Tumor proportion score.*

*Abbreviations: PS=Performance status, Smk=Smoking status, TT=Tumor type, Meta=Metastasis, BM=Brain metastasis, TPS=Tumor proportion score, Tx=Treatment, Mono=Monotherapy, Combo=Combination therapy, M=Male, F=Female, Non-Sq=Non-Squamous.*

## 10.2 PUBLIC DATASET 1 (PD-1): METABRIC BREAST CANCER DATASET

The Molecular Taxonomy of Breast Cancer International Consortium (METABRIC) dataset Evitan (2021) serves as a primary validation source for Our framework. It comprises clinical data from 2,509 unique breast cancer patients, offering a robust real-world cohort for survival analysis.

### 10.2.1 COHORT CHARACTERISTICS

The patient cohort is diverse, with ages at diagnosis ranging from 21.9 to 96.3 years (mean age: 60.4 years). Surgical interventions included both *mastectomy* (complete breast tissue removal) and *breast-conserving surgery* (targeted removal of cancerous tissue). A key strength of this dataset is its representation of rare cancer types; it includes 2,506 breast cancer patients and 3 patients with breast sarcoma, the latter accounting for less than 1% of all breast cancers. The most frequent histological subtype is invasive ductal carcinoma (IDC), with 1,865 occurrences, which aligns with its known prevalence of approximately 80% of all breast cancer diagnoses. These factors collectively affirm that the METABRIC dataset accurately reflects real-world clinical scenarios.

### 10.2.2 PREPROCESSING FOR OUR FRAMEWORK

The dataset contains both numerical and categorical features. To prepare it for Our refinement algorithm (Algorithm 1), we first applied a standardized preprocessing pipeline (Algorithm 2). Furthermore, to capture longer-term survival trends, we extended the analysis threshold from 120 to 480 time units and adjusted the corresponding time bins accordingly.

---

**Algorithm 2** Data Preprocessing Pipeline

---

1: **procedure** PREPROCESS($D, P$)
2:     **for** $f \in$ CategoricalFeatures($D$) $\setminus \{P\}$ **do**
3:         $D \leftarrow$ OneHotEncode($D, f$)
4:     **end for**
5:     **for** $g \in$ NumericalFeatures($D$) **do**
6:         **if** HasMissing($D[g]$) **then**
7:             $D[g] \leftarrow$ FillMissing($D[g], \text{median}(D[g])$)
8:         **end if**
9:     **end for**
10:     **return** $D$
11: **end procedure**

---

## 10.3 PUBLIC DATASET 2 (PD-2): MALIGNANT MELANOMA SURVIVAL DATASET

To further demonstrate the generalizability of Our framework, we incorporated the "Survival from Malignant Melanoma" dataset, available in the `boot` package of the R programming language. This dataset records follow-up data from patients who underwent tumor-removal surgery at the University Hospital of Odense, Denmark, between 1962 and 1977. We acquired the data using the pipeline outlined in Algorithm 3.

### 10.3.1 ENDPOINT DEFINITIONS

A critical aspect of this dataset is the definition of the event of interest. The **'status'** variable indicates the patient's outcome at the end of the study:

- **Status 1:** Died from melanoma.
- **Status 2:** Still alive.
- **Status 3:** Died from causes unrelated to melanoma.

This structure allows for different analytical approaches:

- **Overall Survival:** Event = death from any cause (Status 1 or 3 vs. Status 2).
- **Cause-Specific Survival Analysis:** Event = death from melanoma (Status 1 vs. Status 2 or 3).
- **Competing Risks:** Event = death from melanoma (Status 1), with death from other causes (Status 3) as a competing risk.

For Our study, we focused on the **Overall Survival** approach and we considered death event=1 from any cause or cancer.

### 10.3.2 PREPROCESSING AND ADAPTATION

The **'time'** variable represents the number of days from surgery until either the event (death) or the last follow-up (censoring). For consistency with PD-1 and to improve interpretability, we converted these values from days to months. The 'year' column was excluded from the original dataset to simplify the feature set. Similar to PD-1, we extended the threshold time from 120 to 240 units to accommodate the dataset's timeline and updated the time bins in Algorithm 1 accordingly. Missing numerical values were imputed using the median.

---

**Algorithm 3** Dataset Acquisition Pipeline for Melanoma Data

---

1: **procedure** GETMELANOMADATA
2:      Install and load required R packages: `boot, readr`
3:      Load melanoma dataset: $D \leftarrow$ `boot::melanoma`
4:      Export dataset: `write_csv`$(D, $"melanoma.csv"$)$
5:      **return** $D$                                    ▷ Dataset ready for preprocessing
6: **end procedure**

---

