# OpenReview forum: "SPARC: SURVIVAL PSEUDO-LABEL ADAPTIVE RE- FINEMENT AND CALIBRATION"
_ICLR.cc/2026/Conference — ICLR 2026 Conference Withdrawn Submission_

### Official Review · Reviewer_2Vxa · 2025-10-27

**Soundness:** 3
**Presentation:** 3
**Contribution:** 3
**Rating:** 6
**Confidence:** 3

**Summary:**

This paper presents SPARC (Survival Pseudo-label Regularization and Calibration), a general framework that improves the calibration and robustness of deep survival models. The primary goal is to provide reliable 10-year (120-month) survival estimation, the long-term survival probability, while patients were only followed for less than 74 months, so long-term outcomes remain unobserved, preventing reliable 10-year (120-month) survival estimation. The authors address this gap by introducing a dynamic pseudo-label refinement and calibration framework that transforms incomplete follow-up into extended, biologically consistent survival trajectories.

Overall, SPARC provides a well-motivated and flexible approach for enhancing calibration in deep survival models with a conceptually simple solution that is empirically validated. There are some minor limitations regarding theoretical depth and computational cost.

**Strengths:**

1.  Longterm survival analysis is an underexplored area, this work provides a solution with strong empirical performance.

2.  The pseudo-label regularization integrates easily with many survival models.

3. The methodology is clearly described, and code release is planned.

**Weaknesses:**

1. The pseudo-label process increases training complexity and may require tuning for stability.

2. Theoretical justification for convergence and calibration guarantees could be expanded.

3.  Validation on larger or more heterogeneous datasets would further support generalization claims.

**Questions:**

1.	Predicting long-term survival probability without any observed long-term survivors is a challenging extrapolation problem that requires strong assumptions. Have you performed a sensitivity analysis when the model assumption is violated?

2.	How sensitive is SPARC to the frequency of pseudo-label updates during training?

3.	How does SPARC behave when base model calibration is already strong? Does it still provide additional benefits?

---

### Official Review · Reviewer_TL3y · 2025-10-28

**Soundness:** 2
**Presentation:** 2
**Contribution:** 2
**Rating:** 2
**Confidence:** 3

**Summary:**

This work studies long-term prediction in survival analysis with limited follow-up horizons across multiple datasets. The authors identifies that applying pseudo-labels to survival analysis violates clinical constraints and proposes a method to mitigate these violations. Experimental results demonstrate that the proposed method achieves high predictive performance on survival analysis.

**Strengths:**

- The authors address important problems in survival analysis.
- The paper shows that the proposed method achieves high predictive performance while mitigating the issue that pseudo-labels violate clinical constraints in survival analysis.
- The authors experimentally evaluate their proposed method across multiple datasets.

**Weaknesses:**

- The paper does not provide sufficient justification for using the hybrid Weibull–Kaplan–Meier initialization.
- The paper lacks comparisons to other approaches such as Kaplan-Meier and a static pseudo-label method.
- The paper does not provide sufficient justification for the choice of the proposed blending ratio.
- The paper does not validate the ensemble approach against single models.

**Questions:**

- What is the test set ratio for your dataset, the PD-1, and the PD-2?
- How was chosen the parameter alpha in the PD-1 and the PD-2?
- How was chosen the proposed blending ratio?

---

### Official Review · Reviewer_qyHX · 2025-11-03

**Soundness:** 1
**Presentation:** 1
**Contribution:** 1
**Rating:** 2
**Confidence:** 3

**Summary:**

The paper studies dynamic pseudo-labelling in building survival models with all the necessary constraints for risk models like monotonicity and calibration. The idea is to repeatedly sample pseudo-observations under survival constraints to improve predictive performance.

**Strengths:**

- The paper boasts impressive performance on a few datasets.
- The method is carefully constructed with various sensible constraints on the implications of the pseudo-observations, such as expected survival should be before time of death.

**Weaknesses:**

- Fundamentally, pseudo-observations are supposed to help when you can approximately unbiasedly estimate a target parameter. You seem to be using a different way to compute pseudo-labels that does not compute a leave-one-out estimate. Can you justify  this choice?
- Where are the consistency proofs for the proposed method? Why should we expect this to converge to the truth?
- Evaluations should cover more standard benchmarks. (See more here for example: https://arxiv.org/abs/2101.05346)
- A few choices are not motivated: why only work with the datasets in the paper? Why not use synthetic data to validate the method with ground truth outcomes? Why is your method "long-horizon" when you do not prove anything about extrapolation?

**Questions:**

The authors say that existing methods ask for "independent censoring". How/why does refinement handle dependent censoring? Further, there are non-pseudo-outcome methods that can handle dependent censoring. Are you making the point that you adapted pseudo-observations to handle dependent censoring? How so?


Is the procedure run on the validation data also for the evaluation? How is that reasonable?  The main source of my discomfort with the proposed method is that refining the pseudo-labels to impute predictable times is bound to overestimate predictive performance. To ensure this doesn't happen, I want to see a synthetic experiment where you fundamentally bound the maximum possible concordance (by have the survival time have large variance) and demonstrate that the proposed method does not exceed that performance. Further, why not include a cox regression as a baseline in the experiments?


What do you mean by clinically reliable? Would you say standard cox hazard modeling is not clinically reliable? Or are you saying pseudo-observations aren't clinically reliable?

---

### Official Review · Reviewer_Sfuy · 2025-11-06

**Soundness:** 1
**Presentation:** 1
**Contribution:** 1
**Rating:** 2
**Confidence:** 4

**Summary:**

This paper proposes SPARC, an iterative “dynamic pseudo-label refinement and calibration” framework for long-horizon survival prediction. The paper exam the proposed method on a synthetic Our Dataset and two public datasets.

**Strengths:**

Unfortunately, I don't see much strength in this paper.

**Weaknesses:**

1. The paper has very weak ML/methodological contribution. For an ICLR-style ML venue, the core methodological novelty is quite limited. The paper combines:
   - A simple hybrid pseudo-label initialization using Weibull and KM survival at 120 months;
   - A fairly standard deep neural network architecture with engineered time features and an ensemble of three models;
   - Several ad-hoc iterative tricks (periodic quantile transformation, clipping, moving averages, early stopping, ad-hoc learning-rate decay, periodic model reset, etc.);
   - Survival “constraints” (monotonicity, survival=0 after death) that are conceptually obvious and implemented via post-hoc corrections.

This is essentially an engineering pipeline built from known components, not a new learning principle, not a new loss, not a new architecture, etc. There is no theory (no consistency or convergence analysis, no characterization of what the fixed point of the pseudo-label iteration represents, no error propagation analysis). The “EM-like” iterative flavor is mentioned informally in the discussion but never formalized. As a result, I see no solid methodological contribution that would be of lasting interest to the ML community.

2. Motivation and positioning wrt pseudo-observation literature are weak/misleading. The introduction motivates the method by listing three “limitations” of pseudo-observation approaches: dependence on independent censoring, static pseudo-values, and lack of interpretability/diagnostics. However:
   - Wrt Independent censoring. The paper claims existing pseudo-observation methods crucially rely on independent censoring and thereby suffer in realistic clinical data. But there are already many well-known literatures on pseudo-observations under covariate-dependent or informative censoring, e.g. via IPCW or working models, which is not discussed or cited. The discussion glosses over these developments and gives the reader the impression that pseudo-observation methods are fundamentally tied to naive independent censoring assumptions, which is not accurate.
   - Wrt static vs dynamic labels. The fact that classical pseudo-observations are not iteratively updated is presented as a limitation, but pseudo-values are designed to be asymptotically unbiased estimators of well-defined functionals (e.g., survival probability at a time point, RMST). In many settings, not iteratively changing the labels is a feature: you know exactly what target you are estimating. The paper does not articulate what statistical estimand their dynamic pseudo-labels converge to, if any, and why iterative “refinement” is preferable to a consistent, one-shot pseudo-observation.
   - Wrt interpretability and calibration. The introduction frames pseudo-observations as lacking interpretability and calibration; in practice, many pseudo-observation methods are precisely used because the resulting regression coefficients/outputs are interpretable survival functionals and can be assessed with standard calibration tools. The claimed interpretability of SPARC is never concretely demonstrated -- there are no case studies, no feature importance analyses, no subgroup plots that would justify the interpretability claim.


3. The method is heavily heuristic, with many arbitrary design choices and no justification. Concretely, Algorithm 1 is a long list of heuristic operations:

   - Special treatment of a fixed horizon τ=120 months, chosen because “10-year survival” is clinically relevant, but no quantitative justification is provided, and all modeling is single-time-point anyway.
   - Periodic quantile transformation every 3 iterations, hard clipping of labels to [0.01, 0.99], a weight function inversely proportional to |y−0.5|, periodic full model resets after “stagnation”, blending new predictions with previous pseudo-labels using a 70/30 exponential moving average, gradually tightening constraints up to 95%, etc.
   - The architecture itself (layer sizes, dropout schedule, L2 value, the number of ensemble members, learning rate decay schedule) is also fixed by apparent trial-and-error, with no sensitivity analysis.

None of these choices is theoretically motivated, and the paper does not even provide an empirical ablation study to show which parts are necessary. Without such justification, the method looks like a brittle, hand-tuned recipe rather than a principled algorithm.

4. Evaluation is fundamentally circular and lacks proper baselines. The main dataset (“Our dataset”) has a maximum observed follow-up of 74.7 months, but the model predicts 120-month survival. Since there is no actual 120-month ground truth, the authors evaluate performance almost entirely against their own pseudo-labels: R², MAE, etc. measure self-consistency between the model and the pseudo-labels it iteratively helped to create. This raises several issues:

   - The reported near-perfect metrics on the main dataset are not meaningful in terms of real clinical predictive performance; they show only that the ensemble can approximate its own smoothed targets very well.
   - There is no comparison to standard baselines: e.g., pseudo-observation, pseudo observation for dependent censoring, or even simple pseudo-labeling methods. The introduction criticizes existing methods, but the experiments do not include them, which is unacceptable for a paper claiming methodological contribution.
   - On the two public datasets (METABRIC and the melanoma dataset), results are again phrased in terms of pseudo-labels and internal diagnostics; there is no comparison to the many survival benchmarks already reported on these datasets. Also, calibration results in Figure 1 indicate misclassification on these real datasets.

In summary, the evaluation design does not provide evidence that SPARC is better (or even competitive) with existing survival methods; it mainly demonstrates that a powerful network can fit its own pseudo-labels.

5. The “Our dataset” is reconstructed from KM curves; individual covariates are then simulated to match marginal distributions (median/IQR for continuous variables, proportions for categorical). This causes problems:
   - This procedure does not recover the joint distribution of covariates and outcomes; at best it creates synthetic patients whose marginals mimic published summaries. The relationship between covariates and survival is essentially invented by design choices, not learned from real IPD.
   - Validation of the reconstruction is only briefly described (12-month mortality encoding, KM overlays, RMSE), and it focuses on marginal survival curves, not on the joint structure that matters for individualized prediction.
   - Yet the paper repeatedly talks about “individualized survival modeling” and “patient-level prediction.” Given that the core dataset is synthetic and only loosely tied to real IPD, it is hard to interpret the clinical meaning of the resulting model.

**Questions:**

N/A

**Details Of Ethics Concerns:**

The paper constructs an “Our Dataset” by aggregating 36 published studies. Specifically, the authors "web-scrape" Kaplan–Meier curves from these papers and then reconstruct individual-level time-to-event data from those figures, even though the original datasets are not publicly available. This raises concerns about copyright and terms-of-use compliance for the figures and derived data, which the authors do not acknowledge or discuss anywhere in the manuscript.

---

### Note · Authors · 2026-01-07

I have read and agree with the venue's withdrawal policy on behalf of myself and my co-authors.